# Primary Resistance to Immunotherapy-Based Regimens in First Line Hepatocellular Carcinoma: Perspectives on Jumping the Hurdle

**DOI:** 10.3390/cancers14194896

**Published:** 2022-10-06

**Authors:** Francesca Salani, Virginia Genovesi, Caterina Vivaldi, Valentina Massa, Silvia Cesario, Laura Bernardini, Miriam Caccese, Jessica Graziani, Dario Berra, Lorenzo Fornaro, Gianluca Masi

**Affiliations:** 1Institute of Interdisciplinary Research “Health Science”, Scuola Superiore Sant’Anna, Piazza Martiri della Libertà 33, 56124 Pisa, Italy; 2Unit of Medical Oncology 2, Azienda Ospedaliero-Universitaria Pisana, 56126 Pisa, Italy; 3Department of Translational Research and New Technologies in Medicine and Surgery, University of Pisa, 56126 Pisa, Italy

**Keywords:** immune-checkpoint inhibitors (ICIs), advanced hepatocellular carcinoma (aHCC), first line, primary resistance, primary progressors, TIGIT, LAG3, IL-27, GPC3-CAR T, GPC3-CIRP

## Abstract

**Simple Summary:**

Immune checkpoint inhibitors (ICIs) had been explored extensively in patients affected by unresectable hepatocellular carcinoma. These agents were expected to be the keystones of the disease’s first-line treatment because they were theoretically able to revert the immune suppressive tumor microenvironment of the cancerous liver, and because of their manageable safety profile. However, when used as monotherapies, they showed important activity and efficacy limitations. In this mini-review, we summarize the characteristics of the different ICIs-based regimens which constitute the present gold standard of first-line treatment, then, moving from their shortcomings, we discuss the rationale supporting the strategies currently under investigation: systemic triplets and new paradigms of immune-therapeutic agents such as CAR-T and vaccines.

**Abstract:**

Immune checkpoint inhibitors (ICIs) are a key component of different stages of hepatocellular carcinoma (HCC) treatment, particularly in the first line of treatment. A lesson on the primary resistance which hampers their efficacy and activity was learned from the failure of the trials which tested them as first-line mono-therapies. Despite the combination of anti-PD(L)1 agents with anti-VEGF, anti CTLA4, or TKIs demonstrating relevant improvements in efficacy, the “doublets strategy” still shows room for improvement, due to a limited overall survival benefit and a high rate of progressive disease as best response. In this review, we discuss the results from the currently tested doublet strategies (i.e., atezolizumab+bevacizumab, durvalumab+tremelimumab with a mention to the newly presented ICIs/TKIs combinations), which highlight the need for therapeutic improvement. Furthermore, we examine the rationale and provide an overview of the ongoing trials testing the treatment intensification strategy with triplet drugs: anti-PD1+anti-CTLA4+anti-VEGF/TKIs and anti-PD1+anti-VEGF+alternative immunity targets. Lastly, we report on the alternative strategy to integrate ICIs into the new paradigm of immune therapeutics constituted by CAR-T and anti-cancer vaccines. This review provides up-to-date knowledge of ongoing clinical trials of the aforementioned strategies and critical insight into their mechanistic premises.

## 1. Introduction

Regarding advanced hepatocellular carcinoma (aHCC), the role of immune checkpoint inhibitors (ICIs) have changed. This review aims to summarize and discuss the evolution of ICIs from mono-therapies to therapeutic partners of progressively intensified doublet and triplet regimens. The common thread of these strategies is the attempt to overcome the intrinsic primary resistance that HCC shows against ICIs. However, two upstream factors, other than resistance, limit the feasibility of such treatments: the uneven worldwide access to ICIs due to their high costs, and the contraindication to ICIs constituted by orthotopic liver transplant (OLT). The HCC population undergoing OLT is increasing due to the extension of its indication [1], as is the number of patients experiencing disease recurrence afterward [2]. In an attempt to not exclude these patients from ICIs, their absolute contra-indications in the setting of OLT have been debated. Of the 29 case reports of HCC recurrence treated with salvage ICIs, 68% failed to respond and 32% experienced rejection, even though rejection-specific mortality was far less frequent than cancer-specific one [3]. More promising results in terms of efficacy and safety seem to be provided by ICIs’ employment as down-staging/bridging agents to OLT. Indeed most treated patients experienced a nearly complete response to ICIs neoadjuvant treatment, while very few witnessed reversible non-lethal adverse events [3,4]. Nonetheless, these data are supported by low numerosity observations mainly collected as case reports, thus they might only suggest a change in paradigm from absolute to relative contra-indications.

Supported by the promising results of anti-PD-1 agents at sorafenib failure and by the need for less toxic agents than tyrosine kinase inhibitors (TKIs), nivolumab [5] and pembrolizumab [6,7] were the first ones to be tested as first-line treatments through the CheckMate 459 [8] and Keynote 224 -cohort2 [9] trials, respectively. 

The multicenter, randomized, open-label, international, phase 3 CheckMate 459 trial randomly assigned 743 patients to either sorafenib or nivolumab as first lines choices, with the aim to demonstrate a 26% decrease in the risk of death with the latter. The study’s negative result (HR for OS: 0.85) should be read in the context of key achievements by nivolumab: higher survival rates at landmark time-points (47% vs. 44% at 18 months, 37% vs. 33% at 24 months), greater depth of response (8% difference in objective response rate -ORR), higher dose-intensity (83% vs. 38%), lower dose delays because of treatment emergent toxicities (57% vs. 89%), more favorable physical and functional well-being and longer time-to-deterioration of these indexes. Despite being limited by the low sample size (51 patients enrolled) and the absence of a formal statistical design, the results from cohort 2 of the single-arm open-label phase 2 Keynote 224 drove similar observations. The reported ORR of 16%, the disease control rate of 57%, the median OS of 17 months, and the OS rate at 12 months of 58% supported those of CheckMate 459, respectively equal to 15%, 55%, 16.6 months, and 60%. 

Moving from these results, ICIs mono-therapies have been more recently tested against TKIs as non-inferior alternatives: both durvalumab and tislelizumab proved to represent a competitive strategy to sorafenib in terms of OS. In the newly published randomized Himalaya trial (that will be discussed later), the results of the first-line therapy with the anti-PDL1 durvalumab are in line with those of nivolumab and pembrolizumab [10]. Granting an mOS of 16.56 months, an ORR of 17%, and a DCR of 54.8%, durvalumab proved non-inferior to sorafenib, still not superior. Similarly, the phase III open-label RATIONALE-301 trial [ESMO Congress 2022, LBA36] met the primary endpoint of non-inferiority of tislelizumab versus sorafenib, showing efficacy and activity results consistent with the aforementioned: mOS of 15.9 months, ORR of 14.3% and DCR of 41.8%. As a whole, the observations on nivolumab, durvalumab, and tislelizumab mono-therapies are robustly driven by large study populations (about 700 enrolled patients for each study) and highlight consistent mOS, 2-years OS rates, and ORR results of 16 months, 15% and 39% respectively. Consequently, ICIs monotherapy trials led to their approval as first-line alternatives to anti-angiogenic containing regimens and in Child-Pugh B setting according to NCCN guidelines [11]. The role of ICIs monotherapy in the therapeutic algorithm of aHCC mainly resides in a better safety profile than TKIs. In a phase I/II open-label non-comparative, multi-center trial, 10% of Child-Pugh B aHCC patients treated with nivolumab showed a sustained (>6 months) functional improvement to Child-Pugh A class, likely related to tumor response to the administered ICI [12]. From Himalaya post-hoc analyses, durvalumab with or without tremelimumab showed favorable OS outcome and benefit-risk profiles compared with sorafenib, irrespective of baseline albumin-bilirubin (ALBI) grade [13]. On the other hand, ICIs monotherapy trials shed light on the existence of primary resistance to ICIs in aHCC, thus on the necessity of newer strategies to overcome it. Spy of this phenomenon is the high rate of primary progressors (PP), reported as high as 37% with nivolumab, 33% with pembrolizumab, 45.2% with durvalumab, and 49.4% with tislelizumab. The mechanism of HCC primary resistance to ICIs seems to encompass both tumor intrinsic and micro-environment (TME) levels [14]. Among the former, WNT/CTNNTB1 somatic mutation has the soundest evidence to promote HCC immune escape [15], along with the expression of multiple immune checkpoints such as lymphocyte-activation gene 3 (LAG-3), fibrinogen-like protein1(FGL1), T-cell immunoreceptor with Ig and ITIM domains (TIGIT)-nectin cell adhesion molecule 2 (NECTIN2) and CD 155. Different immune cell populations and epigenetic mechanisms are top players in TME-related resistance to ICIs. Higher CD8+ T lymphocytes, TREM-1+ tumor-associated macrophages, activated hepatic stellate cells and DNA methylation dysregulation have been linked to a more immune tolerant TME [16,17]. 

In this context, recent evidence on the impact of HCC etiology on the immune characterization of TME deserves a mention since it might represent a tailoring tool for ICIs administration in aHCC. Indeed, NASH-induced HCC led to the expansion of exhausted CD8+PD1+ T cells in pre-clinical models and human hepatic biopsies, suggesting the mechanism behind a reduced sensitivity to anti PD(L)1 mono-therapies in this subgroup of patients [18]. A focus on the biological role of some of the most studied ICIs’ resistance determinants is summarized in Table 1 along with their implication in ICIs’ sensitivity.

Given these premises, anti PD(L)1 combination with either anti-angiogenic molecules or different ICIs is being explored as a front-line strategy to overcome monotherapy primary resistance, as discussed below. 

## 2. Doublets: The Current Strategy

In the attempt to improve ICIs’ efficacy in aHCC, three different combination strategies have been so far developed based on the concept of treatment intensification: (a) double ICIs blockade, namely anti PD(L)1/anti CTLA4; (b) anti PD(L)1/anti-VEGF; (c) anti PD(L)1/multi-target tyrosine kinase inhibitors (TKIs). Of these strategies, (a) and (b) have already proven superior to sorafenib, with atezolizumab/bevacizumab representing one of the current standards of care in this setting [29] and tremelimumab/durvalumab (STRIDE regimen) being a more recent valuable competitor [10]. As for (c), reported data are conflicting. If camrelizumab/rivoceranib showed survival benefit over sorafenib supporting another new first-line treatments option for aHCC [LBA35, ESMO 2022], lenvatinib/pembrolizumab [LBA34, ESMO 2022] and cabozantinib/atezolizumab failed to show an overall significant benefit over TKIs alone.

IMbrave and Himalaya were the first phase 3 clinical trials to demonstrate the superiority of a doublet strategy over sorafenib. As a global, open-label study, IMbrave enrolled 501 patients, randomly allocated 2:1 to atezolizumab/bevacizumab or sorafenib, to test the superiority of the experimental arm in terms of OS and progression-free survival (PFS). Both primary analysis and updated results performed beyond expectations: not only efficacy endpoints showed clinically sound improvement with mOS of 19.2 months (HR 0.66) and mPFS of 6.8 months, but the benefit of the strategy was maintained during the follow-up time, with upfront and progressive separation of Kaplan-Meier curves and duration of response >6 months in 87.6%. The activity enhancement was equally remarkable, with 73.6% of DCR, 29.8% of ORR, and 7.7% of complete responses (CR), despite unfavorable prognostic features of the study population, such as macro-vascular invasion of the main portal trunk or the portal vein branch contralateral to the primarily involved lobe, bile duct invasion, or at least 50% hepatic involvement. Quite simultaneously designed, the Himalaya phase 3, global, open-label trial aimed at proving the OS superiority of tremelimumab single-dose priming combined with subsequent durvalumab administration (the so-called STRIDE regimen) over sorafenib, as a primary endpoint. In addition to improved mOS (16.43 vs. 13.77 months, HR 0.78), the key secondary endpoints of durvalumab non-inferiority to sorafenib, prolonged duration of response (65.8% at 12 months), and the increased survival benefit after 9 months of treatment proved the strength of this strategy. As for activity, in contrast to an mPFS superimposed to sorafenib’s one, DCR (60.1%), ORR (20.1%), and CR (3.1%) greatly favored STRIDE. Key differences between the two regimens lay in their safety profile and consequently their feasibility. The addition of an anti-VEGF drug caused 7% upper gastrointestinal bleeding, higher G3-4 hypertension rate (15.2%), and proteinuria (3%), making the evaluation of the presence of gastro-esophageal varices a compulsory up-front screening before treatment administration and a possible limitation of this doublet application. The rate of G3-5 hemorrhagic events in the real-world population is being investigated as the primary endpoint of the phase 3b Amethista trial, whose results will help to tailor patient selection for this regimen [30]. Conversely, the addition of a single anti CTLA4 priming led to an overall lower incidence of any G3-4 treatment-emergent adverse events (50.5% vs. 56.5% with atezolizumab/bevacizumab), but to a higher rate of immune-mediated ones (12.6%), 20.1% of which requiring high-dose steroids. 

Moreover, the influence of HCC etiology on these doublets’ efficacy is an intriguing difference that warrants further confirmation: compared to either HBV- or HCV-related HCC, the non-viral etiology seems to derive less benefit from atezolizumab/bevacizumab, while a greater one with STRIDE, even though no interaction tests were carried out. 

Phase III trials exploring the addition of TKIs to an anti PD(L)1 have been recently reported. In particular, the randomized phase 3, open-label, multicenter COSMIC-312 trial is the first published one to explore the combination of cabozantinib plus atezolizumab over sorafenib, with regard to OS and PFS dual primary endpoints. Despite mOS results from the combination revealing no improvement [31], PFS was significantly improved (6.8 vs. 4.2 months, HR 0.63). Moreover, some interesting observations are prompted by the trial’s results: comparable DCR (78%) and ORR (11%) to the other tested doublets, low rates of PP (14%), the enhanced PFS benefit in the Asiatic and HBV-positive population, and the higher PFS of cabozantinib monotherapy over sorafenib’s one (5.8 vs. 4.3 months) underpinning the contribution of this TKI to the combination’s efficacy. During ESMO 2022 congress, the results of the LEAP 002 and SHR-1210-III-310 trials were presented. Unfortunately, the lenvatinib/pembrolizumab combination did not meet its primary dual endpoint of OS and PFS, despite showing a trend toward improvement over lenvatinib monotherapy. On the contrary, in the SHR-1210-III-310 trial camrelizumab/rivoceranib significantly improved in OS and PFS versus sorafenib. Despite similar endpoint results (21.2 and 221.1 months of mOS for lenvatinib/pembrolizumab and camrelizumab/rivoceranib, respectively), the SHR-1210-III-310 trial was successful, while LEAP-002 trial was statistically negative. Putative contributing factors to such a difference were study design and enrolled population. LEAP-002 trial set lenvatinib plus placebo as a control arm, while the SHR-1210-III-310 trial used sorafenib: using a placebo control arm, the drop-out rate and/or investigator-assessed progression events might have been lowered and both lenvatinib and sorafenib OS outperformed those of the original pivotal trials. In this regard, more stringent eligibility criteria with fitter enrolled patients, improvement of supportive care and earlier initiation of systemic therapy due to multidisciplinary decisions might have improved control arms’ OS. As for the study population, in both trials patients with HBV etiology had better OS: thus, the higher proportion of HBV positivity in the SHR-1210-III-310 trial (75%) might partially explain the differences in outcome between the studies, along with a higher proportion of Asian subjects (83% vs. 31%, respectively). 

Seeking to consolidate the results derived from the aforementioned strategies, test bio-similar compounds, and extend these results to the HBV-positive Chinese population, many clinical trials on doublets are currently ongoing, as reported in Table 2.

Nonetheless, data currently available on doublets suggest that there is room for improvement, especially for what concerns their activity. Indeed, PP rates remain high in these trials. Progressive disease was the best response in 37% and 45.5% in IMbrave 150 and Himalaya respectively; COSMIC-312 and LEAP-002 lowered this value to 14% and 12.2% respectively but failed to translate it into efficacy improvement. Such rates appear to be consistent with the prevalent HBV-positive Chinese setting of ORIENT-32, where 27% PP was described on sintilimab/bevacizumab biosimilar therapy [32]. More encouraging results seem to be provided by the camrelizumab/rivoceranib combination, whose PP rate is 16.2% and which translated into a positive OS endpoint, even though mainly restricted to an Asian (83%) or HBV-positive (75%) population. The importance of this observation resides in the fact that higher rates of PP lead to higher probabilities of hepatic failure and to lower chances of receiving subsequent systemic treatments, ultimately limiting mOS. As a consequence, only 20–40% of patients in ICIs-starting strategies received further regimens, compared analogously to TKIs-starting ones, being 32.6% and 38.7% after lenvatinib and sorafenib respectively [33]. 

Of note, the efficacy of these strategies seems not to be empowered by a more tailored patient selection. ICIs’ efficacy predictive tools borrowed from more immunotherapy-sensitive solid tumors are not extensively investigated in this setting’s clinical trials (with the exception of PD-L1) and, when subgroup data are presented, they do not identify a more susceptible cluster of patients [34] Specifically, microsatellite instability (MSI) status, the most significant ICIs agnostic predictive tool and likely the most meaningful biomarker across gastrointestinal malignancies, is not assessed in the pivotal mono- or combination-therapies aHCC trials. Therefore, we cannot rule out the putative impact of an unidentified proportion of ICIs-sensitive MSI-HCC patients on the efficacy of the tested combinations, despite a reported very low incidence of this condition (<3%) [35,36]. 

Interestingly, subgroup analyses of the doublets’ trials show inconsistent efficacy of different strategies across etiologies, with the anti-angiogenic containing regimens (atezolizumab/bevacizumab, atezolizumab/cabozantinib, lenvatinib/pembrolizumab and camrelizumab/rivoceranib) being seemingly less effective in non-viral etiology than stride regimen.

Consequently, the research strategies to overcome ICI’s primary resistance are heading toward an intensified triplet strategy comprising ICI’s combination with different agents.

## 3. Triplets: A Strategy under Investigation

Triplet systemic regimens under study (Table 3) comprise the combination of: (i) the three already proved-active compounds anti PD1 + anti-CLTA4 + anti-angiogenics (i.e., anti-VEGF or TKIs); (ii) ICIs + chemotherapy (restricted to Asiatic population); (iii) the anti PD1 + anti-VEGF + alternative immunity targets TIGIT, LAG3 or IL-27. As a whole, these strategies aim at targeting simultaneously different pathways which are synergically involved in aHCC pathology. The different strategies’ specific rationales are hereafter recapitulated.

### 3.1. Anti PD1 + Anti CTLA4 + Anti-Angiogenics

The anti PD1 + anti CTLA4 + anti-angiogenic strategy includes either the addition of anti-VEGF or TKI lenvatinib to the double ICIs backbone: each strategy shows rational distinctiveness. 

VEGF blockade addition to an otherwise anti-angiogenic-less treatment addresses the hypoxia-induced hypervascular nature of most HCCs [37,38] and reinforces the suppressive role of CTLA4 blockade on T regulatory lymphocytes (T-reg). Indeed, anti CTLA4 antibodies reduce T-reg mediated suppression of CD8+ T cells in in vitro and in vivo models preventing the internalization of B7 receptors on antigen-presenting cells surface [39,40]. Similarly, anti-VEGF hampers T-reg activity by blocking their VEGFR2 [41] and by reversing MDSCs-dependent T-reg de novo development [42]. An ancillary mechanism that supports the rationale of this triplet is suggested by the immune response elicited by anti-CTLA4 + anti-VEGF doublet in melanoma patients: the increase of cancer-specific targets IgG, such as galectin-1, seen in melanoma patients couldbe translated to the aHCC setting as well [43]. This strategy is further supported by the conflicting results of the recently presented anti PD(L)1/TKIs doublets.

Despite sorafenib being the most studied TKI with regard to this class’s immune-modulatory effects [44], the rationale for combining lenvatinib with the ICIs doublet lies in multiple reasons. Firstly, in the absence of specific contraindications, lenvatinib seems to represent a valid alternative to sorafenib as a first-line TKI choice [45]; secondly, it proved to increase CD8+ activated T lymphocytes in the TME of mice models [46], which are the main players in cancer cytotoxicity. Moreover, recruitment of CD8+ activated T cells to TME by lenvatinib suggests a synergy with the STRIDE regimen which causes an early expansion of the same immune population in peripheral blood, according to Study 22 translational analyses [47]. 

Specifically, NCT04740307 proposes the combination of the co-formulated anti-PD1/anti-CTLA4 MK-1308A (or quavonlimab) with lenvatinib. MK-1308A is a novel humanized immunoglobulin G1 monoclonal antibody that binds to CTLA4 and blocks interaction with its ligands CD80 and CD86. When added to pembrolizumab in the presence of human mixed lymphocyte and monocyte-derived dendritic cells, it increases IFN-Y production by 13 folds [48]. MK-1308A safety and activity when combined with pembrolizumab are being tested in solid tumors by the phase 1/2 MK-1308-001 ongoing multi-cohort study. Results are known for the trial’s arms which enrolled pretreated SCLC patients [49] and first-line NSCLC ones [48]: the combination tested active with an ORR of 18% and 35.1% respectively, seemingly independent of PDL1 cut-offs, and showed a 200-1000 fold increase in circulating Ki67+ CD4+ and CD8+ T cells. These observations grant a rationale for MK-1308A exploration in aHCC, whose susceptibility to ICI is limited by low rates of PD-L1 expression and by the functional exhaustion of Ki67+CD8+ infiltrated cells in the non-viral etiology disease subset.

### 3.2. Anti PD1 + Anti-VEGF + Alternative Immunity Targets

The addition of either LAG3, TIGIT, or IL-27 inhibition to the anti PD(L)1 + anti-VEGF doublet shares a common rationale: all these targets were demonstrated to exert an immune-suppressive effect on TME, thus their inhibition should lead to immunogenicity restoration in HCC. 

LAG3 is a surface receptor of both effectors and inhibitors of immune populations with the function of accelerating T cell exhaustion and blocking T cell proliferation through its canonical (MHC II) and alternative ligands, such as FGL1 [50]. LAG3/FGL1 axis has been identified as the major responsible for LAG3-induced immunity suppression and tumor growth enhancement [51]. Research on LAG3 role in HCC has shown that FGL1 and LAG3 expression is higher in HCC tissues than in normal livers, as opposed to PDL1 and CD8+ cells, and that PDL1^neg^LAG3^high^ HCC cells predict poor prognosis in HCC patients [51]. Being an inhibitor IgG4 monoclonal antibody on the LAG3 signaling pathway, relatlimab combination with nivolumab was recently approved by FDA for advanced unresectable and previously untreated melanoma, thanks to PFS and OS results from Relativity-047 phase 2/3 trial [52]. Moving from these considerations, relatlimab is currently under investigation both in the setting of second-line aHCC [53] and as a peri-operative strategy (NCT04658147), in addition to the first line of the aforementioned trial.

The surface receptor TIGIT has been widely studied in the context of HCC. Mainly expressed by NK, activated and memory T cells and T regs, TIGIT, CD96, and CD226 bind to the common CD155 ligand expressed on many malignant cells and fine-tune functionally opposite signals, with TIGIT exerting inhibition on T lymphocytes [54]. Evidence on TIGIT expression and role in HCC is built on the analysis of immune infiltrate of TME from surgical samples. In a cohort of 47 resected HCC studied for leukocytes infiltration in the tumoral (TILs) and tumor-free regions, TIGIT was enriched in PD1^high^CD8+ TILs and it was co-expressed with LAG3 and TIM3 exhaustion markers. The IFN-y production ability of this population subset was impaired. Double TIGIT and PD1 in vitro blockade enhanced the proliferation of CD8+ isolated TILs significantly more than PD-1 blockade alone with nivolumab. Of note, the co-blockade was able to convert in vitro anti-PD1 non-responders to responders through the functional restoration of CD8+ TILs [55]. As additional pieces of evidence of TIGIT’s role in HCC, TIGIT+CD4+, and TIGIT+T-reg cells were demonstrated to be involved in HCC pathogenesis [56] and TIGIT expression was described as positively correlated to AFP levels [56]. These results hold true also for the HBV-related HCC subset [57].

The role of the immune-suppressive IL-27 in HCC needs to be further dissected. Opposing pre-clinical evidence seems to support the rationale for its successful employment: on the one hand, the downstream pathway of IL-27 receptor demonstrated to activate in-vivo HCC development and to correlate with poor prognosis through the restrain of innate cytotoxic lymphocytes [58], while, on the other hand, it was demonstrated to induce PDL1 expression on different HCC cell lines thus promoting immune-escape [59]. However, it should be acknowledged that PD-L1 up-regulation might serve as a rationale for anti-PDL1 co-targeting, since providing a target otherwise unevenly expressed.

Bavituximab, a monoclonal antibody targeting the phosphatidylserine (PS) located in the inner layer of the phospholipid cell membrane, seems to contribute to anti-cancer response in at least two different ways. On the one hand, it prevents the production of immune suppressive cytokines induced by PS recognition by immune cells; on the other hand, it targets PS expressed on neo-angiogenesis induced endothelial cells, specifically targeting tumor vasculature [60]. Despite the efficacy of bavituximab addition to sorafenib in a phase 2 trial of aHCC was inconclusive, the safety profile was not exacerbated [61] and the experimental drug was eventually tested in combination with pembrolizumab achieving an ORR of 31.3% [62]. The currently investigated strategy of adding bavituximab to a backbone of both anti-angiogenic and anti-PD-L1 drugs might therefore enhance its performance.

## 4. Immunotherapy beyond ICIs: Future Perspectives

Since some of the discussed limitations to ICIs’ efficacy are intrinsic to their mechanism of action, a new paradigm of immunotherapy agents is being explored for aHCC patients. Functionally, they are complementary to ICIs, therefore they provide a sound rationale for being combined with known check-point inhibitors, rather than being tested as monotherapies, as discussed below. An overview of currently ongoing phase I trials of these new strategies is given in Table 4.

### 4.1. Chimeric Antigen Receptor T Cell Therapy

One of the most beaten tracks is the chimeric antigen receptor (CAR) engineered T cell with different potential substrates, such as glypican-3 (GPC3), melanoma antigen gene (MAGE3), human telomerase reverse transcriptase (hTERT), and AFP [63].

GPC3 is a heparan sulfate proteoglycan encoded by the GPC3 gene, approximately expressed in 75% of HCC cells, while not in non-cancer tissues or healthy livers [64]. Evidence on the activity of GPC3-CAR T cells therapy is built on a steadily increasing number of in vitro and in vivo studies [65]. Recent results about patient-derived xenograft (PDX) of HCC engrafted in mice suggested that GPC3-CAR T cells display specific and efficient cytotoxicity against GPC3-positive target cells. Their activity was likely exerted through the up regulation of CD25, CD27, CCR7, and the co-stimulatory receptors CD86 and CD137, which are indicators of enhanced proliferative potential of T cells. Notably, PD-1 and CTLA-4 were also up-regulated. It should be highlighted that GPC3-CAR T cells eradicated tumors from PD-L1 negative-PDXs, while in PD-L1 positive-PDXs GPC3-CAR T cells were less cytotoxic [66]. This observation might grant the rationale for the combination of CAR T cell therapy and ICIs to boost PD-L1-positive HCC clearance, as already reported in breast cancer [67]. Moving from this background, several phase I clinical trials are ongoing, testing GPC3-CAR T cells and other CAR-T therapies in HCC patients.

### 4.2. Vaccines

The rationale for using vaccines against liver cancer resides in the possibility of stimulating the host’s immune system against the tumor. However, it’s now clear that vaccination alone is not able to up-regulate immunity on its own, because of the collateral activation of immune escape mechanisms [68]. Hence, the rationale for combining ICIs with vaccines: (1) in addition to their most studied mechanism of action, ICIs inhibit target tumor cells elimination, thereby enhancing the immunogenicity of vaccines [69]; (2) vaccines stimulate the expansion of the reservoir of effector antigen-specific T-cells [70] and up-regulate the expression of molecules targeted by anti-PD-1 and anti CTLA4, as well as PD-L1 expression on antigen-presenting cells [71], therefore enhancing T cells response. 

An intriguing demonstration by Silva L. et al. illustrates the role of the inflammatory factor cold-inducible RNA-binding protein (CIRP) as a potential subset for vaccination against HCC. CIRP is a human toll-like-receptor-4 (TLR4) ligand produced and released under stress conditions like hemorrhagic shock and sepsis, which triggers inflammatory cytokines production [72]. Once combined with specific peptide antigens, CIRP stimulates CD8-specific responses able to blow out settled neoplastic cells. Tested on immunized mice, the investigators reported that 5 out of 10 mice treated with the combination of CIRP-based vaccine and ICIs bore no more recognizable tumor. In the subsequent step, the authors injected a CIRP-based vaccine containing the human HCC antigen GPC3 (GPC3-CIRP) in mice together with anti-PD-1 and anti-CTLA-4 antibodies. It was confirmed that the combination of the two strategies induced greater responses mainly directed against the 522–530 epitope of HLA-A2*01, leading to the hypothesis that GPC3 harbors CD4 T-cell epitopes that help induce CD8 T-cell responses against 522–530 epitopes [73]. However, despite these encouraging results, weak effects were seen in TILs which maintained an exhausted phenotype, pulling the trigger to future combination strategies and newer targets (Table 4).

## 5. Conclusions

The development of efficacious immunotherapy strategies is of paramount importance in the management of aHCC. Despite the unsatisfying results reported with ICIs monotherapy in the overall population, impressive results in terms of response and disease control are reported in subgroups of patients. Furthermore, the combination of atezolizumab and bevacizumab is currently a widely recognized standard of care as upfront therapy and the new combination of durvalumab and tremelimumab (STRIDE regimen) demonstrated significant improvements compared with sorafenib. Therefore, it is clear that, because of their indirect mechanism of action, ICIs should be combined with other synergistic agents to achieve significant results in terms of activity and efficacy. However, several issues are still open: the right partner for anti-PD(L1) agents among anti-angiogenics, the role of alternative cellular checkpoints, and the feasibility of triplet combinations. Finally, new strategies comprising engineered T cells or antigens will hopefully be developed by the ongoing clinical trials.

## Figures and Tables

**Table 1 cancers-14-04896-t001:** Most-studied determinants of ICIs resistance in HCC.

Determinant	Biological Role	Specificity to HCC	Putative Mechanism of Resistance	Ref.
*WNT/CTNNTB1*	Evolutionary conserved transcriptional pathway, cell-cell adhesion, pivotal hepatic functions since embryonal life.	Its activating mutation relates to over-expression of wnt-target genes, enrichment in beta-catenin, and *PTK2*-related immune exclusion. More represented in viral etiology.	Lower enrichment score of immune signatures: T-cell exclusion, and down-regulation of CCL4. Down-regulation of NKG2D ligand hampering NK-mediated response.	[19,20,21]
LAG-3	Type-I trans-membrane protein acts as a negative immune counterweight during prolonged exposure to tumor antigens and is constitutively expressed by T-regs.	More expressed and more frequently mutated (15%) in HCC tissues than non-malignant livers in TGCA samples; positively correlated with the oncogenic transcription factor E2F1.	High correlation between its expression and immune-suppressive or exhausted tumor environment.	[17,22,23,24]
FGL1	Liver-secreted protein and main functional ligand to LAG-3 inhibiting antigen-specific T cell activation	Significantly down-regulated in HCC samples and correlated with higher grades, presence of metastases and poorer outcomes. FGL1-positive CTC patients showed resistance to ICIs treatment in a limited retrospective case-series	High expression of FGL1 is correlated with higher density of LAG3+: blocking the FGL1/LAG3 can promote T cytotoxicity immunity.	[14,25,26]
TIGIT	Co-inhibitor receptor expressed on T cells and NK, functionally similar to PD-1	Co-factor for T cells functional exhaustion in chronic viral hepatotropic infections; hallmark of immune suppressed HCC TGCA sub-group.	*TIGIT*, *CTLA4* and *ICOS* are co-regulated and co-expressed; TIGIT interaction with NECTIN2 shapes a cancer-promoting immune suppressive environment	[14,27,28]

Abbreviations: LAG3, lymphocyte-activation gene 3; FGL1, fibrinogen-like protein1; TIGIT, T-cell immunoreceptor with Ig and ITIM domains; NK, natural killers; CTLA4, Cytotoxic T-Lymphocyte Antigen 4; ICOS, Inducible T-cell costimulator.

**Table 2 cancers-14-04896-t002:** Ongoing phase II/III trials of the following first-line combination strategies in aHCC setting, registered to Clinical Trial.gov: anti-PD(L)1/anti CTLA4; anti-PD(L)1/anti-VEGF; anti-PD(L)1/multi-target TKIs.

Strategy	Trial Name and/or CTC Identification	Recruitment	Phase	Comparator	Interventions ^targets^
Anti-PD(L)1/anti CTLA4	NCT04720716	China	III	Sorafenib ^a^	IBI-310 ^b^ (ipilimumab bio-similar) + sintilimab ^c^
	NCT04039607/ CheckMate 9DW	global	III	Sorafenib or Lenvatinib ^d^	Ipilimumab ^b^ + nivolumab ^c^
Anti-PD(L)1/anti-VEGF	NCT04605796	China	II	-	JS001, toripalimab ^c^ + bevacizumab ^e^
	NCT04560894	China	II/III	Sorafenib ^a^	SCT-I10A^c^ + SCT510 (bevacizumab bio-similar) ^e^
	NCT04741165	China	II	-	HX008 ^c^ + bevacizumab ^e^
	NCT03973112, arm IV	China	II	-	HLX10, serplulimab ^c^ + HLX04 (bevacizumab biosimilar) ^e^
Anti PD(L)1/TKIs	NCT04443309	China	I/II	-	Camrelizumab ^c^ + lenvatinib ^d^
	NCT04401800	China	II	-	Tislelizumab ^c^ + Lenvatinib ^d^
	NCT04183088	Taiwan	II	Regorafenib ^f^	tislelizumab ^c^ + regorafenib ^f^
	NCT04523493	global	III	Lenvatinib ^d^	JS001, toripalimab ^c^ + lenvatinib ^d^
	NCT03841201/IMMUNIB	Germany	II	-	Nivolumab ^c^ + lenvatinib ^d^
	NCT04741165	China	II	-	HX008 ^c^ + lenvatinib ^d^
	NCT05441475, part b	China	II	-	Atezolizumab ^g^ + ABSK-011 ^h^
	NCT03439891	USA	II	-	nivolumab ^c^ + sorafenib ^a^
	NCT04443322	China	II	-	Durvalumab ^h^ + lenvatinib ^d^

Interventions’ targets: ^a^ BRAF, VEGFR1-3, FLT3, PDGFR-beta, FGFR1, RET, KIT; ^b^ CTLA4; ^c^ PD-1; ^d^ VEGFR1-3, FGFR1-4, PDGFR alfa, RET, KIT; ^e^ VEGF-A; ^f^ VEGFR1- 3, KIT, PDGFR alfa, PDGFR beta, FGFR 1-2, angiopoietin receptor, BRAF, MAPK 11, FRK, ABL1, RET; ^g^ PD-L1; ^h^ FGFR4.

**Table 3 cancers-14-04896-t003:** Ongoing clinical trials of the triplet systemic strategy as first line for aHCC.

Trial Identification	Study Phase	Treatment Arms	Targets	Primary End-Point
NCT05363722	Ib	IBI 310 (ipilimumab biosimilar) + bevacizumab + sintilimab	anti PD1 + anti CTLA4 + anti-VEGF	ORR
NCT04740307 MK-1308A-004	II	pembrolizumab/quavonlimab (MK-1308A) + lenvatinib	Coformulated anti PD1/anti CTLA4 + TKI	DLTs ORR
NCT05363722	III	Camrelizumab + Folfox4	anti PD1 + chemotherapy	OS
		Camrelizumab + placebo		
NCT04948697 AdvanTIG-206	II	ociperlimab + tislelizumab + BAT1706	Anti-TIGIT + anti PD1 + anti- VEGF	ORR
		tislelizumab + BAT1706	anti PD1 + anti- VEGF	ORR
NCT05337137 Relativity-106	I/II	Relatlimab + Nivolumab + Bevacizumab	Anti-LAG + anti PD1 + anti-VEGF	DLTs ad PFS
		Placebo + Nivolumab + Bevacizumab	anti PD1 + anti-VEGF	
NCT05359861	II	SRF388 + Atezolizumab + bevacizumab	Anti-IL27 + anti PDL1 + anti-VEGF	PFS
		Placebo + Atezolizumab + bevacizumab	anti PDL1 + anti-VEGF	
NCT05249569	II	Axitinib + Avelumab + Bavituximab	Anti-VEGFR + anti PDL1 + anti-phosphatidylserine	RR

Abbreviations: ORR, objective response rate; DLTs, dose limiting toxicities; OS, overall survival; PFS, progression free survival; RR, response rate.

**Table 4 cancers-14-04896-t004:** Ongoing clinical trials of CAR-T and vaccine for aHCC.

Treatment Strategy	Trial Identification	Study Phase	Treatment Arms	Primary End-Point
CAR-T cells therapy	NCT02905188 GLYCAR trial	I	GPC3-CAR (GLYCAR T cells) + lymphodepleting chemotherapy (Cyclophosphamide and Fludarabine)	DLT
	NCT03884751	I	CAR-GPC3 T Cells	DLT + MTD
	NCT03980288	I	CAR-GPC3 T Cells (in part II: combination with TKI or anti- PD(L)1)	DLT + MTD
	NCT04121273	I	CAR-GPC3 T Cells	DLT
	NCT03993743	I	CD147-CART hepatic artery infusion	AEs
Vaccine + RFA/surgery	NCT03067493 RAMEC trial	II	RFA or surgery +/− Neo-MASCT	DFS + immune response rate
Vaccine + ICIs	NCT04248569	I	DNAJB1-PRKACA peptide vaccine + Nivolumab + Ipilimumab	AEs + change in INF-producing DNAJB1-PRKACA-specific CD8/CD4 T cells

Abbreviations: DLT, dose limiting toxicity; MTD, maximum tolerated dose; AEs, adverse events.

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
