# Peer review of "Primary Resistance to Immunotherapy-Based Regimens in First Line Hepatocellular Carcinoma: Perspectives on Jumping the Hurdle"

_cancers, 2022, doi:10.3390/cancers14194896_

Round 1

Reviewer 1 Report

I would like to commend this paper for its exhaustive, excellent review on the topic of CPI-treatment in HCC. I only have very few points to suggest. Specifically, I suggest that:

-          That issue of a potential proportion of MSI-high or mismatch repair deficiency subpopulation of HCC patients is briefly addressed

-          The new data from ESMO should be added (as surely the authors intend to do)

-          I would avoid stating too categorically that Lenvatinib overtook for good the place of Sorafenib.

-          While discussing the rationale for triple treatment I suggest that the negative results of TKI-based combination suggest that 2-CPI+VEGF should be the next rationale choice

-          To this regard I suggest that the MONTBLANC Study (2022-001201-48) is added to the list of triplet-treatment trials. This study is the first of its kind to combine Durvalumab, Tremelimumab and Bevacizumab and is due to begin recruitment soon.

Author Response

Response to Reviewer 1 Comments

I would like to commend this paper for its exhaustive, excellent review on the topic of CPI-treatment in HCC. I only have very few points to suggest. Specifically, I suggest that:

(1) That issue of a potential proportion of MSI-high or mismatch repair deficiency subpopulation of HCC patients is briefly addressed

(2) The new data from ESMO should be added (as surely the authors intend to do)

(3) I would avoid stating too categorically that Lenvatinib overtook for good the place of Sorafenib.

(4) While discussing the rationale for triple treatment I suggest that the negative results of TKI-based combination suggest that 2-CPI+VEGF should be the next rationale choice

(5) To this regard I suggest that the MONTBLANC Study (2022-001201-48) is added to the list of triplet-treatment trials. This study is the first of its kind to combine Durvalumab, Tremelimumab and Bevacizumab and is due to begin recruitment soon.

Firstly, we would like to thank the reviewer for his/her positive feedback on our paper and for the suggestions given to further improve it.

Response (1): We acknowledge the necessity to briefly address this observation, therefore we added the following sentences “Moreover, the efficacy of these strategies seems not to be empowered by a more tailored patients selection. In fact, ICIs’ efficacy predictive tools borrowed from more immunotherapy-sensitive solid tumours are not extensively investigated in this setting’s clinical trials (with the exception of PD-L1) and, when subgroup data are presented, they do not identify a more susceptible cluster of patients [21]. Specifically, micro-satellite instability (MSI) status, the most significant ICIs agnostic predictive tool and likely the most meaningful biomarker across gastrointestinal malignancies, is not assessed in the pivotal mono- or combination-therapies aHCC trials. Therefore, we cannot rule out the putative impact of an unidentified proportion of ICIs-sensitive MSI-HCC patients on the efficacy of the tested combinations, despite a reported very low incidence of this condition (<3%) [22] [23]”

Response (2): Yes, we intended to, since data presented at ESMO are pivotal to the discussion of this subject. The manuscript was prepared before ESMO 2022 took place, when only the Companies’ press releases were available. The text has been amended accordingly (track change modality and yellow highlight) to include data from LEAP-002, SHR-1210-III-310 and RATIONALE-301 trials and their data have been critically discussed.

Response (3): We acknowledge that the sentence “in the absence of specific contraindications, lenvatinib has overtaken sorafenib as gold standard first-line TKI” is a bold statement; so we have modified it into “Firstly, in the absence of specific contraindications, lenvatinib seems to represent a valid alternative to sorafenib as first-line TKI choice ”

Response (4): As for point (2), all data regarding new ESMO results on TKI-ICI doublets have been added along with a comprehensive discussion of these results. Moreover, the sentence “This strategy is further supported by the conflicting results of the recently presented anti PD(L)1/TKIs doublets” has been added.

Response (5): We are willing to add this study to table 2 to make it more comprehensive. However, we were not able to identify the study on clinical trials.gov. Could the reviewer please grant us with the information needed to comprise it?

Reviewer 2 Report

This article was aimed to review the status of use of immune checkpoint inhibitors (ICIs) in unresectable advanced hepatocellular carcinoma(aHCC). Some concerns of this article are listed below.

1.As we known, the role of ICIs in treatment of aHCC is still limited due to the unsatisfactory efficacy and economic burden of patients compared. Therefore, as this article entitled” Primary resistance to immunotherapy-based regimens in first line hepatocellular carcinoma: perspectives on jumping the hurdle”. The possible mechanisms of primary resistance of ICIs regimen in treating aHCC and how do the ongoing trials of combinations of different ICIs overcome this drug resistance should be reviewed extensively in this article.

  Thus, the discussion in line 66 to 86 should be extended and authors please provide additional figures and tables to summarize the suggested mechanisms of drug resistance of ICIs and the rationale of choice of different combination of ICIs in strategies in treating aHCC. This would allow an easier flow of ideas making the manuscript more reader-friendly.

  2.Currently, the use of ICIs in patients with recurrence of HCC after liver transplantation is still controversial. Authors may add some paragraphs to review this topic. 

Author Response

Response to Reviewer 2 Comments

This article was aimed to review the status of use of immune checkpoint inhibitors (ICIs) in unresectable advanced hepatocellular carcinoma (aHCC). Some concerns of this article are listed below.

  1. As we known, the role of ICIs in treatment of aHCC is still limited due to the unsatisfactory efficacy and economic burden of patients compared. Therefore, as this article entitled” Primary resistance to immunotherapy-based regimens in first line hepatocellular carcinoma: perspectives on jumping the hurdle”. The possible mechanisms of primary resistance of ICIs regimen in treating aHCC and how do the ongoing trials of combinations of different ICIs overcome this drug resistance should be reviewed extensively in this article. Thus, the discussion in line 66 to 86 should be extended and authors please provide additional figures and tables to summarize the suggested mechanisms of drug resistance of ICIs and the rationale of choice of different combination of ICIs in strategies in treating aHCC. This would allow an easier flow of ideas making the manuscript more reader-friendly.

2.Currently, the use of ICIs in patients with recurrence of HCC after liver transplantation is still controversial. Authors may add some paragraphs to review this topic.

Firstly, we would like to thank the reviewer for pointing out these criticisms which are giving us the possibility to improve the paper.

Response (1): We agree with the reviewer that at the current state of art, ICIs role in the therapeutic landscape of aHCC still needs refinement and that ICIs feasibility is limited by still unsatisfactory efficacy results, high costs and uneven access in different geographical regions. However, the latter point is not comprised in the discussion since in most cases the regulatory agencies had considered accelerated approvals to these compounds, and the pharmaceutical companies had frequently granted temporary access to these strategies while waiting for the definitive approval. The review was intended to summarize the journey of ICIs as first line options, from the monotherapy strategy to progressively intensified combinations, through the most important clinical trials of this setting; the detailed mechanistic description of resistance was not its main topic, since it had already been extensively discussed in other recent review papers. The “rationale of choice of different combination of ICIs in strategies in treating aHCC” is already given as premise in all the following paragraphs: 3.1, 3.2, 4.1, 4.2, therefore we have not added further details with this regards. On the other hand, we acknowledge that the resistance determinants enlisted in line 66-86 could have been more clearly summarized, thus a table summarizing the biological role of the most studied determinants, their specific implication in HCC and the putative mechanisms of ICIs resistance was added (see Table1).

Response (2): We acknowledge that liver transplant still represents at least a relative contraindication to ICIs treatment, thus, in a way, a limitation to this strategy, even though it cannot be considered as a proper mechanism of resistance to it. However, we have briefly addressed this topic in introduction, as suggested by the reviewer.

Round 2

Reviewer 2 Report

none